# Socioeconomic Status, Health and Lifestyle Settings as Psychosocial Risk Factors for Road Crashes in Young People: Assessing the Colombian Case

**DOI:** 10.3390/ijerph18030886

**Published:** 2021-01-20

**Authors:** Andrea Serge, Johana Quiroz Montoya, Francisco Alonso, Luis Montoro

**Affiliations:** 1DATS (Development and Advising in Traffic Safety) Research Group, INTRAS (Research Institute on Traffic and Road Safety), University of Valencia, 46022 Valencia, Spain; 2Dipartimento Scienze Statistiche, Faculty: Ingegneria Dell’informazione, Informatica e Statistica, Sapienza Università di Roma, 00185 Rome, Italy; jkquirozm@unal.edu.co; 3FACTHUM.Lab (Human Factor and Road Safety) Research Group, INTRAS (Research Institute on Traffic and Road Safety), University of Valencia, 46022 Valencia, Spain; luis.montoro@uv.es

**Keywords:** epidemiology, socioeconomic status, traffic safety, public health

## Abstract

The social determinants of health influence both psychosocial risks and protective factors, especially in high-demanding contexts, such as the mobility of drivers and non-drivers. Recent evidence suggests that exploring socioeconomic status (SES), health and lifestyle-related factors might contribute to a better understanding of road traffic crashes (RTCs). Thus, the aim of this study was to construct indices for the assessment of crash rates and mobility patterns among young Colombians who live in the central region of the country. The specific objectives were developing SES, health and lifestyle indices, and assessing the self-reported RTCs and mobility features depending on these indices. A sample of 561 subjects participated in this cross-sectional study. Through a reduction approach of Principal Component Analysis (PCA), three indices were constructed. Mean and frequency differences were contrasted for the self-reported mobility, crash rates, age, and gender. As a result, SES, health and lifestyle indices explained between 56.3–67.9% of the total variance. Drivers and pedestrians who suffered crashes had higher SES. A healthier lifestyle is associated with cycling, but also with suffering more bike crashes; drivers and those reporting traffic crashes have shown greater psychosocial and lifestyle-related risk factors. Regarding gender differences, men are more likely to engage in road activities, as well as to suffer more RTCs. On the other hand, women present lower healthy lifestyle-related indices and a less active implication in mobility. Protective factors such as a high SES and a healthier lifestyle are associated with RTCs suffered by young Colombian road users. Given the differences found in this regard, a gender perspective for understanding RTCs and mobility is highly suggestible, considering that socio-economic gaps seem to differentially affect mobility and crash-related patterns.

## 1. Introduction

Much research has been conducted in the field of traffic safety, since the consequences of Road Traffic Crashes (RTCs) and Road Traffic Injuries (RTIs) have been recognized as a major concern for public health [1,2]. The numbers show that, worldwide, around 1.35 millions of people die and 50 million are injured as a consequence of road traffic crash-related events [3]. Unfortunately, despite the efforts made by different researchers, governments and institutions, these occurrences are still present in our life.

An important case study is the one corresponding to developing countries. These countries are especially impacted, since, in addition to being economically affected by low economic growth, they also suffer a huge number of crashes. The worrisome part of this phenomenon is that several studies point out how RTCs seem to be following a tendency to increase [4,5], affecting specific population groups such as younger and vulnerable road users. The presence of RTCs is detrimental to both the economic wellbeing and the macroeconomic performance of countries [6]. RTCs and RTIs keep a tight relation with the economy and health, not only in what concerns the consequences and burdens associated with their occurrence and prevention [6,7], but also in the explanation of their causes through mobility, as well as in what concerns the psychosocial risk and/or protection factors, such as health determinants and lifestyle.

RTCs as events are predictable, and they can be approached from a perspective that works directly with road actors, or with what is known as the human factor [8,9]. The importance of studying them is rooted in that this factor seems to be determinant in RTCs, with at least 67% of crashes resulting from human errors [10], a proportion that varies depending on the studied population. Traditionally, the study of the human factor in road safety has focused on drivers, since they are the ones with access to vehicles, which may be the element that is most related to traffic. Numerous studies have demonstrated the important role of drivers in the occurrence of RTCs [11,12]; however, nowadays experts have centered their attention on other groups of road actors, especially those that are considered vulnerable, such as pedestrians, cyclists and children. This has led to research on human factors going beyond the study of human errors associated with the driving task [13].

Additionally, this element highlights the need of studying crash rates from a psychological and social perspective, which has already demonstrated how the interrelation of society–economy–health can lead to specific groups being more vulnerable to suffering RTCs, or even influence the perception that people have of these events [14].

Therefore, the study and understanding of crash occurrences, aiming at their possible prevention, must take into account other types of elements beyond the crash itself, and from a multi-disciplinary approach [1]. Let us start with the consideration of socioeconomic status (SES): there is a wide research corpus that has demonstrated how the socioeconomic possibilities of people can influence their life, or even determine their opportunities in the world before they are even born [15]. This happens not only on an individual level, but it also influences different dimensions of life within societies [16].

The SES is framed and recognized as a determinant of health, as well as a predictor of different conditions and situations [17,18], both positively and negatively [19]. The SES widely influences the health risk, having an impact on people’s healthcare, environment, and psychosocial functioning [20]. The evidence points out that SES can be used as an explicative element of RTCs, and there is research proving that belonging to a lower social class is associated with a higher frequency of suffering deathly traffic crashes [21,22]. At the same time, an average or high SES appears to be more related to an intention or attitude towards risky behaviors [23], and people who engage in risky behaviors have more chances of suffering a crash [24,25].

As a concept, SES is the result of many other variables, such as age, sex, neighborhood and country [26,27,28,29]; for its understanding, Socioeconomic Position (SEP) indicators must be taken into account as well, referring to social and economic factors influencing what position an individual will have within a society [29]. Monthly salary and wealth stand out particularly [30,31], together with residence and housing [29,32], education [33,34,35] and occupation [36,37].

In a similar way, health is a major dimension that must be considered. As it has been said, RTCs represent a health burden related to deaths, and they are also an important cause of living with an injury-related disability [1,38]. In addition to this, it has been found that the driving task, if prolonged, has a negative impact on health, for instance through an increase of stress and fatigue [39]. The acquisition of not-so-healthy habits, such as sleeping less and performing less physical activity, has also been reported [40]. Moreover, mental health can be compromised too, and health status concerns are associated with poor driving behavior [41]. For what concerns the physical aspect, some argue that people with high body mass may be at higher risk of suffering a RTC [42,43].

To sum up, SES, health and lifestyle influence, and even determine, the psychological and social risk or protective factors: better SES, health and lifestyle indices are associated with a better psychological health [44], and the poorer the psychological health, the more probabilities of suffering RTCs [45,46]. Now, in order to study the relation of these topics with traffic and road safety, all the above should be considered, starting from the following premises: which country are we talking about? What are the characteristics of people at risk, and of those who suffer these crashes? In the case of Colombia, it has been reported that a driver can be four times more likely to die in a crash, compared with a driver in Spain [47], in addition to a ratio of 18.5 RTC deaths every 100,000 inhabitants [3]. Moreover, young Colombians are a risk group, and they are vulnerable to RTCs [48].

Taking into account that the consideration of SES, health and lifestyle allows us to understand why traffic crashes seem to be present and possibly increasing in developing countries, the objective of this work is to construct indices related to these major topics in order to explore their relationship with RTCs and mobility in a sample of young Colombian participants. The null hypothesis that there are no significant differences between groups is going to be tested for each index, expecting that the groups with the most vulnerable SES, unfavorable health and worse lifestyle will present more crashes and will have patterns of more active mobility. As specific objectives, the study aims at: (1) developing SES, health and lifestyle indices for this country, that will take into account sociodemographic variables, SEP indicators and health-related information; (2) assessing self-reported RTCs and mobility features depending on the SES, health and lifestyle indices.

## 2. Materials and Methods

### 2.1. Participants

Colombia is a country with 44.164 million inhabitants [49]. Several studies have pointed out that, taking into account a confidence level of at least 95% and a 5% margin of error, a minimum sample size of *n* = 385 is required in order to conduct meaningful analyses [50,51,52]. We will take this number as our sample reference, assuming that a population group adequately represents the population from which such group is extracted [53,54]. According to the Statuary Law, from 1855 and from 2018, as a modification of the Young Citizens Status, in Colombia young people are those with an age ranging from 14 to 28 years old, and youth is considered the stage during which one’s intellectual, moral, physical, economic, social and cultural autonomy are being built [55]. It is reported that young people represent at least 21.8% of the country’s total population [56].

Following a cross-sectional design, the sample was collected through convenience sampling, and participants who were older than 17 were included. Since young people were the study’s target, the research relied on the cooperation of university lecturers, who emailed their contacts an invitation to participate. Overall, 20 professors were invited, and 15 of them accepted, thus having a 75% margin of acceptance. A total of 731 interviews were completed, and after a cleaning and refining process through the age filter > 17 and < 29, a final sample of *n* = 561 was selected, therefore reducing the margin of error to 4.14%. Most of the respondents (65.95%) were from Bogotá, the most populated city in the country, and from municipalities from Cundinamarca surrounding the capital (30.12%).

### 2.2. Procedure and Data Analysis

Facing the limitations of web-based surveys but highlighting their economic advantages, their efficiency in collecting data, their reduction of interviewer biases [57], and the fact that, through a rigorous design and development, “results from an online survey may be no different than paper based survey results” [58], this study gathered the data using an online survey named “Encuesta de Salud y Seguridad Vial” (“Survey on Road Safety and Health”), whose average completion time was 40 min. This survey collected data on sociodemographic and crash records information, as well as on some specific scales. It was reviewed by two experts: a psychologist with traffic safety experience, and a civil engineer with experience in the assessment of human factor in transportation. After their recommendations, the instrument was tested in a pilot study including 50 participants, which allowed for the elimination of ambiguous items.

To achieve the general and specific objectives, Principal Component Analysis (PCA) was used to construct the indices. Chi Square Independency Test and Student’s *t*-test for Independent Samples were performed to compare group means, both with a 95% level confidence, testing the null hypothesis that there are no significant differences between groups. The *p* values were adjusted through False Discovery Rate (FDR), which is thought to be the best approach, “as it not only reduces false positives, but also minimizes false negatives” [59]. Finally, a violin plot to show the full distribution of the data was charted. All the previous steps were performed using the free software environment for statistical computing and graphics R [60].

### 2.3. Index Construction

Broadly speaking, an index is a measure composed of other variables that allows for the representation of a construct or result [61]; it can be used as a quantitative indicator of the researched idea. Indices can be developed in different ways, however, in the case of SES constructs and health-related indices, the PCA is a variable reduction approach which remains constantly used and is thought to be useful in epidemiological studies, despite its limitations. Howe, Hargreaves and Huttly [62] consider that a PCA “involves replacing a set of correlated variables with a set of uncorrelated ‘principal components’ which represent unobserved characteristics of the population.” Additionally, beyond the method that is used, what will weight on the results of the model seems to be the categorization of the variables [62]. This perspective was taken into account to construct three (3) indices, considering that every time an item is categorized differently, the PCA results change; thus, a total number of 76 items, contemplating the original item and its different forms of categorization, were considered (See Appendix A).
For what concerns SES: Socio-economic stratification, which in Colombia is a way to classify the residential properties that must receive public services and subsidies according to their social stratum, are established in the Law 142 from 1994 [63].SEP indicators include the wage reported in the Minimum Legal Wages for the year 2020 in Colombia, the occupational status and the educational level.Evaluation of wealth assets: residing in one’s own house (belonging to the individual or to the nucleus of co-habitation, where no rent is to be paid); access to a computer; money for leisure; savings; debts; permanent access to the internet; and covered month (which means the feeling of being able to manage with the available monthly income).Number of people who inhabit the home. The average number for Colombian homes is 3.3 in urban zones and 3.9 in rural zones. Furthermore, 52.7% of homes with 5 or more people reported incomes below 2 minimum wages [64]. This type of family structure, or cultures that foster familistic societies, can be not so good on an economic level. This is due to the fact that, regardless of the possible social support that these networks provide, economic resources seem to be more associated with living alone instead [65].Regarding health: the perception of having a good health, the use of medicines and the body mass index (BMI) were evaluated. In addition, some of the main causes of death and non-communicable diseases were considered as well: cancer, diabetes, hypertension/high blood pressure, dyslipidemia (evaluated through the vector: HDL-LDL cholesterol, triglycerides) and cardiovascular diseases. Additionally, diagnosis of a mental/psychological disorder, general self-reported stress and fatigue were taken into account.For lifestyle: having a sedentary life; doing sports at least 3 times a week; doing sports at least 30 min every time; smoking; drinking alcohol; self-assessment of one’s eating habits; walking; and using a bike were considered.Sleeping hours per day (24 h). Regularly sleeping less than 7 h per night can lead to adverse health conditions, such as weight gain and obesity, hypertension, depression, diabetes, heart disease and stroke, and increased risk of death; between 7 and 9 h could be considered a normal range for young adults and adults, while more than 9 h could be enough for young adults and for people recovering from sleep debt or suffering from illnesses. Nevertheless, it is still unknown whether sleeping more than 9 h per night could imply health risks [66].

Additionally, for contrasting the utility of the indices, the following variables were taken into account too:RTCs in a dichotomous way No/Yes (0–1): have you ever suffered a traffic crash? Suffering a crash as a road actor, a variable that was considered when the participant was matched in the vector: having a traffic crash, or a crash as a passenger, on a bike, as a pedestrian or as a driver. The variables that compose this vector were also used to study the contrasts.RTCs as continuous variable: number of traffic crashes throughout one’s life; number of crashes suffered as a passenger in one’s life; number of crashes suffered on a bike; number of crashes suffered as a pedestrian; number of crashes suffered as a driver during one’s life.Age and sex. As several road traffic studies have demonstrated, there are significant differences in the traffic crashes suffered by different age groups [67], by men and women [67], as well as various concerns in the economic and health-related fields [31].

### 2.4. Compliance with Ethical Standards

The present study obtained its ethical approval from the Research Ethics Committee of the University Research Institute on Traffic and Road Safety at the University of Valencia (IRB: E0002080419). Additionally, it complied with the guidelines established by the Code of Ethics and Bioethics of Psychologists [68]. Following this code, participants completed the survey only if they had previously agreed with an informed consent form that emphasized confidentiality and data protection rights, with special attention to the fact that the data would be used only for research purposes, thus encouraging participants to provide sincere answers.

## 3. Results

With these data, the descriptive analyses used to understand the participants’ profiles were performed according to sex and income, and they can be consulted in Table 1. In total, 413 women (73.88%) and 146 men (26.12%) participated, and their mean (SD) age was 20.83 (2.49) years. In total, 59.3% of the sample reported having finished their high school studies; Status 3—middle (40.4%), Status 2—low (41.1%) and Status 1—low-low (7.5%), represents 89.05% of the total sample.

### 3.1. PCA Indices Construction

Variables accounting to equal or more than 95% in any of the answer categories were discharged. To construct the PCA indices, the subset of variables was scaled, allowing for the use of covariances matrices. The Kaiser-Meyer-Olkin (KMO) factor adequacy was tested to be higher than 0.5, which is considered acceptable for employing the selected method. Several models were tested, considering 70 possible variables. These were reduced according to their contribution to the final models and to the cluster explaining the possible components, in addition to the related theory. The final components manage to explain around 56.3% and 67.9% of the total variance, and they were used to generate three indices: SES, Health and Lifestyle. The respective loadings with an absolute cutoff of |0.34| for components with eigenvalues ≥1 are displayed in Table 2. Missing data were omitted in the final model in order not to affect its predictive value (see Table 2).

The indices were constructed through the sum and ponderation of the variance explained by each eigenvalue ≥ 1, to be then re-scaled within a 0–1 range. For SES and Lifestyle indices, a value equal to 1 corresponds to the most favorable socioeconomic status and to the best lifestyle conditions, respectively. For what concerns the health index, 0 represents a lack of unfavorable health conditions and 1 represents the presence of illness. Some works suggest considering only the first component of the PCA (Comp.1) to construct the indices, and, therefore, the Comp.1 of each model was tested in contrast with another index equivalent to the sum and ponderation of all components with eigenvalues ≥ 1. However, the relations explained only by the Comp.1 were not found to provide better or worse contrast results, which is why we chose, as final indices, those that ponder components in order to increase the variance explained by the model. The indices were also categorized in terciles that were Low (<0.43), Average (0.43–0.61), and High (>0.61) in the case of SES; for what concerns the Lifestyle, they were Unhealthy (<0.44), Average (0.44–0.81), Healthy (>0.81); and regarding Health, they corresponded to Good Health (>0.28), Average Health (0.20–0.28), and Poor Health (<0.20).

### 3.2. Means and Frequency Contrast

To explore the behavior of the indices categorized in terciles, the Chi-square test of Independence was employed and reported, together with the adjusted standardized residuals, where values higher than 1.96 indicate more cases than expected, while values lower than 1.96 indicate fewer cases than expected. The effect size is reported through the contingency coefficient (see Table 3). To begin with the SES index, statistically significant differences were found in the driving task. It is attention-worthy how there are more cases than expected presenting a high SES in the case of those who drive. On the other hand, suffering a crash as a pedestrian presents differences as well; specifically, people with a higher SES report more crashes like these, while an average SES implies fewer people who have suffered a crash as pedestrians. For what concerns the Health Index, no significant differences were found.

On the other hand, the Lifestyle index shows differences in comparison with the Health index, since the adjusted standardized residuals show more cases of poor health than expected in the case of the unhealthy lifestyle group; also, there were fewer cases of poor health in the healthy lifestyle group. Differences in the use of bikes show that there are fewer cases of people not using bikes in the healthy lifestyle group. Regarding bike crashes, more cases than expected were found in the healthy lifestyle too, for those who were involved in this type of crash. The self-reported crashes also showed significant differences: there were more cases than expected when considering unhealthy lifestyles. The lifestyle index also presented differences with the crashes suffered as a driver, finding more cases than expected in the unhealthy lifestyle category and in those who suffered the crash, and fewer cases in the healthy lifestyle group. Finally, differences were found in the sex variable, too: there are fewer women with a healthy lifestyle in comparison with the group of men (see Table 3).

For what concerns the continuous variables, Student’s *t*-test for independent samples was also tested (see Table 4), considering crash rates and mobility as contrasting variables. For what concerns the SES index, it was found that those who drive presented an average SES higher than those who do not. There are no mean differences related to the Health index. Regarding the lifestyle index, it was found that those who reported suffering a crash had a lower lifestyle mean; those who suffered crashes as drivers also presented a lower mean; and, finally, those who rode a bike had a lifestyle mean that was higher than those who did not.

Finally, Figure 1 shows a violin plot for the indices that display the variables’ distribution depending on the reported crashes and on the sex variables (as an example of the possible distributions the indices could have across the participants’ features). The figure allows us to visualize the predictive power of the indices, observing that the lifestyle index is the one adjusting to the data curve in the most adequate way.

## 4. Discussion

Understanding that developing countries are severely affected by RTCs and that this issue must be approached from a multi-dimension and interdisciplinary perspective, this work has proposed the need of studying crash rates and mobility patterns in young Colombians through SES, health and lifestyle as predictors of psychosocial risk factors. To our knowledge, this is the first study of this type that was ever performed in Colombia. By means of a reductive approach and to explain between 56.3% and 67.9% of the variance, three indices were constructed: SES, Health and Lifestyle, since the evidence appoints them as determinant elements to be considered when comprehending who suffers RTCs and why.

To begin with, the variables reduction led us to discharge a total of 54 variables, leaving three models composed of 10 variables (SES), 7 variables (Health) and 5 variables (Lifestyle). This reduction also allowed for a better understanding of how, despite the fact that there are variables highlighted in other countries that we expected would be valuable in these models too (such as the number of people in the home, or the hours of sleep), this did not apply to the population of young Colombians, emphasizing the idea that it is necessary to perform studies focused on the specific issues of each country [69].

### 4.1. Mobility and RTCs Patterns of Young Colombians

To begin with, this study points out some interesting mobility patterns. The majority of young people report walking in their city (93.76%), but their participation as road actors starts to decrease with the use of vehicles; only 26.38% of them use a bike, and even fewer drive a motor vehicle (12.3%), mostly male drivers. Overall, 17.29% of them report that they have been involved in a traffic crash at least once in their life. However, the proportion of those who have been involved in a crash, regardless of their road role, increased up to 39.9%: this leads us to acknowledge that, as other authors have already pointed out [48], young people are indeed at risk for dangerous situations on the road. Additionally, in both cases men reported a higher number of crashes, following the gender-related tendencies associated with RTCs [70,71].

### 4.2. Social and Health Determinants in Young Colombians’ RTCs

#### 4.2.1. Socioeconomic status (SES) and Young Colombians

SES is a determinant of health, as well as of the risky and/or protective actions that a person performs when living. Vulnerable SES and health imply severe detriments for the individual’s quality of life, and the proportion of these inequalities are highly present in developing countries. The problem is that, as some studies have pointed out, the more crashes happen, the bigger the social and economic burden becomes for a country [9,72]. A heavier burden probably corresponds to a lower investment in the development of laws and in the work on road safety, which is a reason why, in addition to the deaths associated with this phenomenon, we are facing a political and economic issue that negatively feeds back on itself. It is not a surprise that vulnerable subjects could be more involved in crashes in countries with poor or still-developing policies, as we were able to verify with this work.

It was found that the indicators of young Colombians were associated with detrimental social conditions. Following the Colombian socio-economic classification, around 48.65% of participants are below the 3rd (middle-low) status, and the debt variable had a considerable weight on the third PCA component. However, the educational factor, among others, was slightly higher than expected in this population, counterbalancing the model so that the index’s terciles point out groups that are more or less similar. This is probably due to the participants mostly living in the country’s capital, and to them being financially supported by their families [73], a support that could also have an influence on the health status through the reduction of psychological stressors [74]. However, this variable did not have any weight on the SES model.

Generally speaking, this index highlighted interesting relations (though fewer than expected) contrasting with variables related to young drivers. To begin with, it was found that those who drive are more associated with high SES, and the index mean is higher for them than it is for those who do not drive. This could be explained by the fact that driving allows the person to move more easily in the city, or even to work more easily, and that, of course, having access to a vehicle is linked to an economy that accumulates capital [75]. As we have said before, the driving task is different depending on sex: men drive more, and those who drive report higher salaries.

On the other hand, those who have experienced crashes as a pedestrian are in the high SES tercile. This provides evidence to reject our initial hypothesis, in which we considered that high SES would present fewer relations with crashes, which is a source of concern not only considering that pedestrians are the most vulnerable road actors [76], but also because, according to the theory, high SES should correspond to a protective factor. In this case, beyond the SES the road safety conditions of Colombia should be taken into account, in addition to the alarming death and injury rates of RTCs and the walkability perception [77].

#### 4.2.2. Health, Lifestyle and Young Colombians

Moving on to the health index, it did not show significant contrasts in the present analysis. However, we can notice in Figure 1, in the part addressing the contrast with crashes, how the proposed model includes the majority of the cloud data within its distribution. The non-existence of significant relations is not necessarily a reason for discharging the construct of RTCs’ study: on the contrary, we believe that the results are caused by the population being young, and by the prevalence of illnesses being quite low, as it can be seen in the index’s terciles. In addition, research on young drivers’ health is more related to their tendency to drink and consume substances [78], which corresponds rather to the field of healthy lifestyle habits (without being excluded from the health sphere).

Actually, it was found that the lifestyle index presents differences from the health index, and there are more cases than expected presenting a poor health in the high/good lifestyle category. However, the relations between this index and mobility seem to be more important (assuming that young people do not get sick so often). To begin with, those who use the bike have a healthy lifestyle and represent a higher proportion of the highest index. This result is important in terms of sustainable mobility, but it also represents benefits for physical [79] and mental health, and it can even have therapeutic effects on some specific populations [80,81]. Nevertheless, it was found that people with a healthier lifestyle suffer more crashes when riding a bike. This is quite concerning, since the message of mobility in the country’s context would then be against the promotion of health. As Evans states: “many people say they would cycle more if the roads were safer—the biggest deterrent to more cycling is high traffic speeds and volumes. There is obviously a vicious circle to be reversed here” [79]. Even so, the study of cyclists’ behavior must be deepened, since they are road actors too, and they could contribute greatly to the occurrence of crashes.

On the other hand, and complementarily, it was found that those reporting that they have suffered RTCs have an unhealthier lifestyle, and, in addition, drivers also have one of the unhealthiest lifestyles. This result is consistent with the findings of other countries and age groups, where it was concluded that driving can even be considered a sedentary activity: driving versus walking [82]. As a sedentary activity, driving can lead to unhealthy habits that are then quite difficult to change [83], with undesirable effects in the short- and long-term.

Finally, the sex and age variables showed important differences that, as we have seen, mark some of the patterns of SES and mobility. It was also found that men are those who keep the healthiest life habits in comparison to women, as other works have shown [84], in addition to having a higher mean of bike crashes. Clearly, we have some risky dynamics at play for young males in Colombia, associated with crash rates. However, this applies to women too, especially in terms of their lower participation in road life and their less healthy life habits. Clearly, a gender perspective must be taken into account in order for women to become more active mobility agents, and for men to be less prone to suffer RTCs. For what concerns age, groups older than 21 engage with the road more, they drive more, they use bikes more, but they also suffer more crashes.

As our results suggest, the work that must be carried out in the country is deep. Joining the same call for action as other authors in what concerns youth [85,86], protecting young Colombians from RTCs must be a priority. It is essential to ensure that they have favorable socioeconomic and psychosocial conditions for their development as well, always following a gender perspective. On the other hand, being active in mobility cannot be a synonym of suffering crashes. If a country aims at enhancing mobility and fostering the use of alternative transport means, such as bikes, it must protect its road actors and provide them with safe contexts so that people will take on an active mobility role through the care of health [87].

## 5. Conclusions

As a result of this research, we now know that SES, Health and Lifestyle as constructs follow a special cluster in the case of young Colombians, and variables highlighted in other countries were not significant in the case of this population. Moreover, sensitive socioeconomic conditions are quite common in this country, and there is a situation of social and economic vulnerability for young people, who, interestingly enough, present high levels of education.

One of the achievements of this study was the construction of three models that allow for the generation of SES, Health and Lifestyle indices in the population of young Colombians, which provide information on the crashes and mobility patterns they have, as well as on differences between groups. The main findings were: (1) drivers are associated with higher SES and driving. The action of driving is associated with higher incomes. High SES is not necessarily associated with protection, since pedestrians belonging to this group report higher crash rates; (2) the prevalence of illnesses is low, and it does not affect mobility or crash rates in this population; (3) people with a better lifestyles use bikes more and report more crashes when using them. Unhealthier lifestyles are associated with more RTCs, and with the driving task; (4) sex and age do establish SES, lifestyle and mobility patterns. Men keep healthier life habits than women, they drive more, they use the bike more, but they also report more crashes than women. Women participate less in the road life, and they have less healthy habits. Finally, the results allow us to draw the conclusion that protective factors such as a high SES and a healthier lifestyle are associated with RTCs in this population, and the age group over 21 engages with the road more, they drive more, they use the bike more, but they also suffer more crashes.

Finally, even though some results may seem obvious, they had not been reported yet; and this is a payoff when working on the RTCs prevention of young Colombians. Additionally, we hope that this work will leave the readers with more questions than answers, and, thinking of the results, we would like to draw the attention to the following interrogatives: is not encouraging people to have a better lifestyle through exercise the objective of health prevention policies? Is not leading us to more sustainable and equal cities the objective of mobility? Then why should taking care of one’s health and cities end up being a risk for young people? The work that is left now consists of further researching the population of drivers and non-drivers in order to answer these questions.

### Limitations and Future Research

Despite the efforts that were made, the analyses we performed could present limitations, and the existence of confounding variables must be evaluated through other methods. Moreover, the size of the sample must be increased for future applications, not only to widen the number of participants, but also to include people from other geographical places in the country, which would diminish the limitations when researching a developing country [69]. In addition to this, future studies will need to obtain more funding, with the aim of performing samplings that are proportional to age, gender and road users (specially drivers).

Even though the constructed indices present an acceptable percentage of the explained variance, the construction and proposal of models that may explain the SES, health and lifestyle associated with young Colombians with more power must be fostered. In addition to this, other conceptual models for the construction of indices should be considered, for instance, a cumulative proposal instead of a reductive one for the model construction [88].

We hope that the results of this work will be useful for understanding the dynamics associated with RTCs in a developing country, and, moreover, with a population group that is at risk. The work of variable reduction that we have performed can be useful for future studies so it they will reduce the application time in what concerns the sociodemographic variables and allow for focus on deepening the researched topics. Additionally, these indices can be extended to the research of other issues since their construction does not depend on mobility or crash variables, but rather used them for contrast. Regarding future research associated with this study, it is worth highlighting the necessity to improve the health index and its predictive value. In future works, we hope to collect data that go beyond self-reports, especially in what concerns health factors, through quick check-ups and revisions of the participants’ medical history, together with visiting the participants’ homes in order to contrast the socioeconomic information. We hope to do this with at least one subsample, considering the economic and ethical implications. Finally, it would be useful to consider the severity and nature of RCTs for implementing specific prevention strategies.

## Figures and Tables

**Figure 1 ijerph-18-00886-f001:**
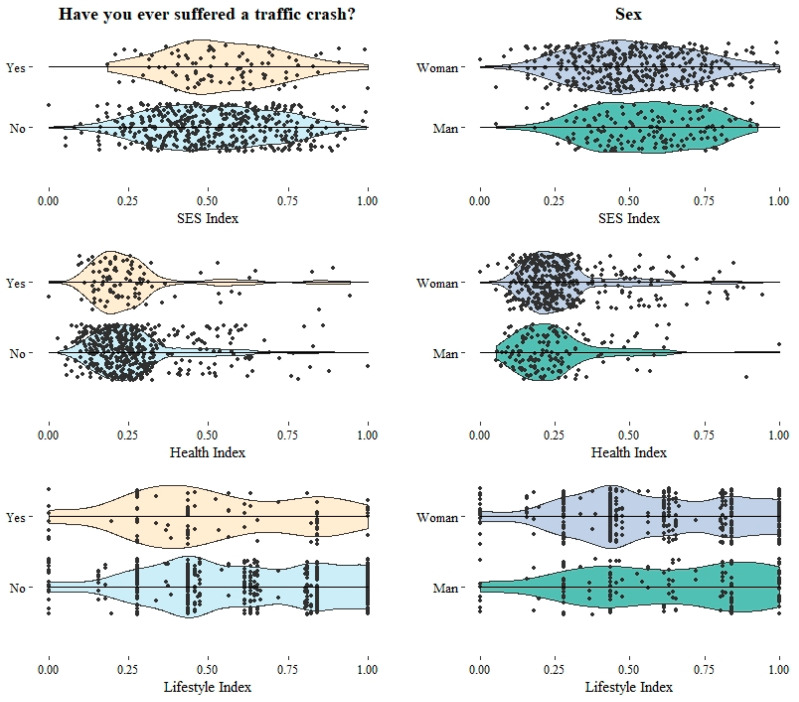
Violin plot of the indices for reported crashes and sex.

**Table 1 ijerph-18-00886-t001:** Sociodemographic characteristics and information as road actors, crossed by sex and income.

Variable Mean (SD)	Fr	Sex		Income SMLMV			
		Man % (*n* = 146)	Woman % (*n* = 413)	None (*n* = 160)	<1 (*n* = 263)	1–2 (*n* = 111)	>2 (*n* = 27)
Age		χ^2^ = 11.645, *p* = 0.009, C = 0.143		χ^2^ = 98.227, *p* < 0.001, C = 0.386			
20.83(2.49)		21.4(2.6)	20.63(2.43)	19.99(2.13)	20.58(2.09)	22.23(2.76)	22.63(3.64)
18	94	11 a	18.9 b	29.4 b	12.9 a	7.2 a	18.5
19–21	284	47.3	51.8	48.8	60.5 b	36.9 a	22.2 a
22–24	126	26	21.1	18.8	20.5	33.3 b	18.5
25–28	57	15.8 b	8.2 a	3.1 a	6.1 a	22.5 b	40.7 b
Educational level				χ^2^ = 22.572, *p* = 0.007, C = 0.197			
Primary school or lower	2	0	0.5	0.6	0	0.9	0
High school or technical	334	52.7	62	61.9	54 a	73.9 b	40.7 a
University	220	46.6	36.6	36.9	45.2 b	24.3 a	55.6
Postgraduate or PhD	5	0.7	1	0.6	0.8 a	0.9	3.7
Socioeconomic stratification				χ^2^ = 32.525, *p* < 0.001, C = 0.235			
Status 1 low-low	42	8.2	7.3	4.4	10.3 b	6.4	3.7
Status 2 low	229	38.4	42.1	44.9	35.5 a	55.5 b	14.8 a
Status 3 middle	225	39	40.8	40.5	41.2	34.5	55.6
Status 4 or higher	61	14.4	9.8	10.1	13	3.6 a	25.9 b
Occupational situation				χ^2^ = 100.112, *p* < 0.001, C = 0.389			
Unemployed or studying only	355	61	64.3	90.6 b	61.8	35.1 a	33.3 a
Employed	205	39	35.7	9.4 a	38.2	64.9 b	66.7 b
Do you drive any type of motor vehicle?		χ^2^ = 10.327, *p* = 0.001, C = 0.135		χ^2^ = 9.002, *p* = 0.029, C = 0.126			
No	492	80.1 a	90.3 b	90.6	88.2	86.5	70.4 a
Yes	69	19.9 b	9.7 a	9.4	11.8	13.5	29.6 b
Do you walk in your city?							
No	35	7.5	5.8	8.8	5.7	3.6	7.4
Yes	526	92.5	94.2	91.2	94.3	96.4	92.6
Do you use a bike in your city?		χ^2^ = 33.055, *p* < 0.001, C = 0.236					
No	413	55.5 a	79.9 b	80.6	71.1	71.2	66.7
Yes	148	44.5 b	20.1 a	19.4	28.9	28.8	33.3
General reported crashes							
No	464	77.4	84.7	88.1	80.2	81.1	81.5
Yes	97	22.6	15.3	11.9	19.8	18.9	18.5
Crashes reported		χ^2^ = 14.654, *p* = 0.002, C = 0.160					
0.29(0.79)		0.49(1.15)	0.22(0.58)	0.18(0.55)	0.32(0.82)	0.36(0.95)	0.37(0.84)
None	464	77.4 a	84.7 b	88.1	80.2	81.1	81.5
1 acc	59	10.3	10.7	6.9	12.9	11.7	3.7
2 acc	20	4.8	3.1	3.8	3.4	1.8	11.1
3 or more acc	18	7.5 b	1.5 a	1.2	3.4	5.4	3.7
Crashes as a road actor		χ^2^ = 18.492, *p* < 0.001, C = 0.179					
No	340	45.9 a	66.1 b	68.1	57	61.3	48.1
Yes	221	54.1 b	33.9 a	31.9	43	38.7	51.9

Notations used at the table. SD: Standard deviation; Fr: Frequency; SMLMV: Minimum legal wages in Colombia for the year 2020; χ^2^: Chi square, *p*: *p*-value, C: contingency coefficient; a: Corrected typified residue < −1.96; b: Corrected typified residue > 1.96.

**Table 2 ijerph-18-00886-t002:** Principal Component Analysis for three indices: Socioeconomic, Health and Lifestyle.

Variable	Comp.1	Comp.2	Comp.3	Comp.4	Comp.5
Socioeconomic status SES (*n* = 556)					
Occupational situation (does not work/student-works)	0.32	**0.53**	0.03	0.01	0.01
Socioeconomic stratification (low-low, low, middle, high)	0.31	−0.27	−0.14	**0.34**	**0.59**
Educational level (low, intermediate, high, high-high)	0.09	−0.17	**0.66**	**0.46**	−0.31
Income (continuous in Colombian pesos)	**0.34**	**0.37**	−0.06	**0.41**	−0.14
Residing in one’s own house (No/Yes)	0.14	−0.17	**−0.7**	0.33	**−0.45**
Having access to a computer (No/Yes)	**0.35**	−0.07	0.08	**−0.49**	**−0.49**
Money for leisure (No/Yes)	**0.44**	−0.14	0.21	0.03	0.11
Having debts (reversed No/Yes)	−0.15	**−0.57**	0.02	0.06	−0.13
Access to the internet (No/Yes)	**0.38**	−0.16	−0.08	**−0.39**	0.25
Covered month (No/Yes)	**0.42**	−0.26	0.01	−0.04	−0.08
Eigenvalue	**1.88**	**1.63**	**1.08**	**1.03**	0.85
Proportion of variance	18.87%	16.29%	10.78%	10.30%	8.52%
Cumulative variance	18.87%	35.16%	45.95%	56%	64.78%
Health (*n* = 557)					
BMI (continuous in kg/mts^2^)	0.04	**0.55**	0.15	**0.58**	**0.57**
Hypertension (No/Yes matched in the vector: hypertension and high blood pressure)	−0.01	**0.55**	0.09	**−0.75**	0.24
Dyslipidemia (No/Yes matched in the vector: cholesterol, LDL, HDL, triglycerides)	−0.18	**0.53**	0.2	0.21	**−0.76**
Diagnosis of a mental/psychological disorder (No/Yes)	0.27	0.32	**−0.65**	−0.11	−0.12
Perception of good health (No/Yes)	**0.41**	0.04	**−0.51**	0.21	−0.05
General stress (assumed as continuous 0–10)	**0.61**	0.05	0.33	−0.04	−0.11
General fatigue (assumed as continuous 0–10)	**0.6**	−0.09	**0.37**	−0.05	−0.07
Eigenvalue	**1.81**	**1.23**	**1.07**	0.93	0.85
Proportion of variance	25.92%	17.59%	15.33%	13.31%	12.16%
Cumulative variance	25.92%	43.51%	58.84%	72.15%	84.31%
Lifestyle (*n* = 561)					
Having a sedentary life (reversed No/Yes)	**0.57**	0.02	0.08	**0.66**	**0.48**
Exercising 3 times per week (No/Yes)	**0.58**	0.12	0.03	0.07	**−0.8**
Exercising for 30 min every time (No/Yes)	**0.56**	0.09	0.02	**−0.74**	**0.36**
Smoking (reversed No-ex/Yes)	0.14	**−0.67**	**−0.72**	0	−0.03
Drinking alcohol (reversed No-ex/Yes)	0.05	**−0.72**	**0.68**	−0.07	−0.05
Eigenvalue	**2.19**	**1.2**	0.78	0.47	0.36
Proportion of variance	43.88%	24.06%	15.54%	93.60%	71.60%
Cumulative variance	43.88%	67.95%	83.49%	92.84%	100%

Loadings with an absolute cutoff of |0.34| are shown in bold.

**Table 3 ijerph-18-00886-t003:** Results for chi-square test of Independence.

Variable Mean(SD)	Fr	SES Index 0.52(0.19)			Health Index 0.27(0.15)			Lifestyle Index 0.59(0.28)		
		Low (*n* = 186)	Average (*n* = 183)	High (*n* = 187)	Good (*n* = 187)	Average (*n* = 186)	Poor (*n* = 184)	Unhealthy (*n* = 127)	Average (*n* = 269)	Healthy (*n* = 165)
Health Index								χ^2^ = 15.081, *p* = 0.005, C = 0.162		
Good	187	23.6	33.1	42.1	23.6	33.1	42.1	23.6 a	33.1	42.1 b
Average	186	33.1	33.5	33.5	33.1	33.5	33.5	33.1	33.5	33.5
Poor	184	43.3	33.5	24.4	43.3	33.5	24.4	43.3 b	33.5	24.4 a
Drive any type of motor vehicle		χ^2^ = 7.569, *p* = 0.023, C = 0.116								
No	492	91.4	89.1	82.4 a	87.2	88.7	87.5	88.2	88.5	86.1
Yes	69	8.6	10.9	17.6 b	12.8	11.3	12.5	11.8	11.5	13.9
Do you walk in your city?										
No	35	6.5	5.5	7	7	4.8	6.5	7.1	6.7	4.8
Yes	526	93.5	94.5	93	93	95.2	93.5	92.9	93.3	95.2
Do you use a bike in your city?								χ^2^ = 18.778, *p* < 0.001, C = 0.180		
No	413	73.7	76.5	72.2	70.1	74.2	76.1	77.2	79.6	61.2 a
Yes	148	26.3	23.5	27.8	29.9	25.8	23.9	22.8	20.4	38.8 b
Reported crashes								χ^2^ = 8.866, *p* = 0.012, C = 0.125		
No	464	87.1	79.2	81.3	80.2	83.3	84.8	74 a	85.9	84.2
Yes	97	12.9	20.8	18.7	19.8	16.7	15.2	26 b	14.1	15.8
Crashes riding a bike								χ2 = 11.228, *p* = 0.004, C = 0.140		
No	487	87.6	89.6	82.9	84.5	85.5	90.2	89.8	90 a	79.4 a
Yes	74	12.4	10.4	17.1	15.5	14.5	9.8	10.2	10 b	20.6 b
Crash as a pedestrian		χ^2^ = 10.322, *p* = 0.006, C = 0.169								
No	281	79.7	86.2 b	68.2 a	79.3	77.6	80.3	75.4	77	84.5
Yes	74	20.3	13.8 a	31.8 b	20.7	22.4	19.7	24.6	23	15.5
Crash as a driver								χ^2^ = 11.804, *p* = 0.003, C = 0.382		
No	45	68.8	45	75.8	58.3	71.4	65.2	40 a	58.1	91.3 b
Yes	24	31.2	55	24.2	41.7	28.6	34.8	60 b	41.9	8.7 a
Sex								χ^2^ = 6.567, *p* = 0.037, C = 0.108		
Man	146	22.6	26.5	29.9	31.6	25.8	21.4	27	21.6 a	32.7 b
Woman	413	77.4	73.5	70.1	68.4	74.2	78.6	73	78.4 b	67.3 a
Age										
18	94	16.7	15.3	17.1	20.3	18.3	11.4	13.4	18.2	17
19–21	284	50	50.8	51.3	47.1	50	54.9	57.5	47.6	50.3
22–24	126	23.7	21.9	22.5	24.1	22.6	21.2	23.6	21.2	23.6
25–28	57	9.7	12	9.1	8.6	9.1	12.5	5.5	13	9.1

Notations used at the table. SD: Standard deviation; Fr: Frequency; χ^2^: Chi square, *p*: *p*-value, C: contingency coefficient; a Corrected typified residue < 1.96; b Corrected typified residue > 16.

**Table 4 ijerph-18-00886-t004:** Mean comparisons for independent samples.

Contrasting Variable	Continuous	Mean No/Man	Mean Yes/Woman	t.test	df	C.low	C.high	*p*	*p*.ad	EF
Reported crashes (No = 464, Yes = 97)	Lifestyle Index	0.60	0.52	2.69	139.83	0.02	0.14	0.008	0.027	0.08
	Age	20.69	21.53	−2.93	134.97	−1.40	−0.27	<0.001	0.015	−0.84
Crash as a road actor (No = 340, Yes = 221)	Age	20.53	21.30	−3.54	447.00	−1.19	−0.34	<0.001	<0.001	−0.77
Drive any type of motor vehicle (No = 492, Yes = 69)	Crash as a driver	0.22	0.84	−4.73	68.00	−0.95	−0.39	<0.001	<0.001	−0.67
	Age	20.67	21.99	−3.73	82.99	−2.01	−0.61	<0.001	0.002	−1.31
	Income in SMLMV	0.08	0.23	−2.94	75.71	−0.26	−0.05	0.004	0.017	−0.15
	SES Index	0.51	0.58	−2.39	83.23	−0.12	−0.01	0.019	0.049	−0.06
Crash as a driver (No = 45, Yes = 24)	Lifestyle Index	0.66	0.43	3.31	55.93	0.09	0.36	0.002	0.009	0.22
Using a bike in the city (No = 413, Yes = 148)	Age	20.58	21.54	−3.92	242.39	−1.44	−0.48	<0.001	0.001	−0.96
	Lifestyle Index	0.57	0.65	−3.16	261.10	−0.14	−0.03	0.002	0.011	−0.08
Sex (Man = 146, Woman = 413)	Crash riding a bike	0.65	0.15	4.18	173.96	0.27	0.74	0.000	0.001	0.50
	Reported crashes	2.18	1.41	2.79	171.50	0.08	0.47	0.006	0.038	0.28
	Age	21.40	20.63	3.11	240.01	0.28	1.25	0.002	0.019	0.77

Notation *t*-test: T statistic; df: Degree of freedom; C.low: confidence interval low; C.high: confidence interval high; *p*: *p*-value, *p*.ad: *p* value adjusted; EF: effect size.

## Data Availability

The datasets and code used and/or analyzed in the present study are available from the corresponding author on reasonable request.

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
