# Peer review of "Socioeconomic Status, Health and Lifestyle Settings as Psychosocial Risk Factors for Road Crashes in Young People: Assessing the Colombian Case"

_ijerph, 2021, doi:10.3390/ijerph18030886_

Round 1
Reviewer 1 Report
The manuscript presents an interesting study investigating the impacts of socioeconomic status, health and lifestyle factors and their association with road traffic crashes amongst a cohort of young Columbian students.
Overall, the methodology is appropriate and the results are of interest to the reader. I would like the author to include an appendix of the 70 items included in the PCA analysis. This will improve the repeatability of the study design.
My primary concern with the manuscript is the quality of writing and the authors need to undertake further proof reading of the manuscript before it is suitable for publication. Some examples include the inconsistent use of "life-style" throughout the manuscript. Use of acronyms that are not presented in the manuscript (RCAs) and some poorly written sentences (e.g. line 97 "four more probabilities" & Line 112 "44,164,417 millions in habitants" (that is 44,164 billion)) and generally there are some grammatical and punctuation errors.
Notwithstanding these issues, with careful review and some rewriting by the authors I see no major reason why the manuscript would not be suitable for publication.
Author Response
Dear reviewer, we appreciate your comments, and we are glad that you found value in our research. You will find our responses to your comments at the attached document.

Reviewer 2 Report
This submission deals with a potentially relevant topic, the impact of SES on road crashes. The writing must be improved and the submission would benefit greatly from the services of a skilled technical editor.
My cocerns are listed below:
- Here are just a few examples of problems with the writing:
- Line 49: RCA – has this acronym been defined at its first use?
- Line 59: ‘remarks’ does not seem to be the correct word in this sentence AND ‘crashes rates’ should be ‘crash rates’, “Additionally, this element remarks the need of studying crashes rates from a psychological and social perspective, that has already demonstrated how the interrelation of society economy-health can lead to specific groups being more vulnerable to suffering RTCs, or even influence the perception of people on these events“
- Line 112: “Colombia is a country with 44,164,417 millions of inhabitants [47], which, considering a confidence level of 95% and a 5% margin of error, requires a minimum sample size of n=385 in order to conduct meaningful analysis [48].”
- I would like to see further details on how this number was calculated
- Line 124: “29<” should be “<29”
- Line 211: “the 3rd status, middle low or lower, represents 89.05% of the total sample.”
- I don’t know what this means.
- Table 1 – if I’m interpreting these data correctly, approximately 88% of the sample are non-drivers, so concluding anything about factors causing crashes may be spurious, at best - this is the central problem with this submission
- another concern is that all data are self-reported; while such is a good way to collect the thoughts, opinions, and feelings, it is far less desireable to collect "factual" information via this method
- Line 255: “the adjusted standardized residuals show more cases of poor health than expected, in the case of the unhealthy life-style group; also, there were fewer cases of poor health in the healthy life-style ” – these outcomes seems self-evident and not requiring of statistical analysis or even reporting
- Discussion – gender and crashes are hopelessly confounded w/ the sampling flaws (vast majority of sample is non-drivers and majority of drivers is male)
- Line 301: “However, the proportion of those who have been involved in a crash, regardless of their road role, increases up to 39.9%: this leads us to acknowledge that, as other authors have already pointed out [46], young people are indeed at risk for dangerous situations on the road.” – the authors give no indication of the severity of these crashes or whether participants had the opportunity to indicate the nature or severity of any reported crash events
- Line 356: “Nevertheless, it was also found that people with a healthier life-style suffer more crashes when riding a bike.” Again, this sounds like circular reasoning. Presumably, bicycling is one factor which places a person in the healthier lifestyle, thus one would naturally expect this group to suffer more bicycle crashes
- Line 376: “For what concerns age, groups older than 21 engage with the road more, they drive more, they use bikes, but they also suffer more crashes.” Not surprising that people who are on the road are the ones suffering traffic crashees
Author Response
Dear reviewer, we appreciate your comments, and we are glad that you found value in our research. Please find attached our responses together with a new version of the manuscript.

Reviewer 3 Report
The article presents an interesting study especially for a deeper reflection on sustainability in cities and opens the opportunity to be able to be replicated in more different populations besides young people. Of course, the fact that it is mostly a study with young people, must be properly safeguarded.We believe that it has a good framework and adequate methodology.
When we suggest correction of some minor errors, we mean that it should be reviewed carefully as there are some grammatical and punctuation errors that can help to improve the article. We still consider that the summary should be clearer, namely with respect to lines 21 and 22, about the objectives of the study where it should be more explicit in the explanation. They should also mention that the sample collection focuses on Bogotá and Cundinamarca and not all Colombia.
We also suggest that references should be globally revised in their reference form: e.g. Lines 560, 563 and 570 and 589.
Author Response
Dear reviewer, we appreciate your comments, and we are glad that you found value in our research and methodology. Please find attached our responses together with a new version of the manuscript.

Round 2
Reviewer 2 Report
I believe the authors have done a good job of responding to the various comments, and I believe the submission is now almost ready for publication. However, there are still some writing issues and typos that should be addressed - I've included a few below.
- Line 42- “their related injuries (RTIs)”
- If you’re defining an acronym, you must define it – not accomplished here – presumably you mean Road Traffic Injuries…?
- Line 44 – “millions are injured as a consequence of road traffic crashes-related events”
- “crashes-related” should be “crash-related”
- Line 58 – “The importance of studying them is rooted in that this factors seems to be determinant in RTCs, with at least 67% of crashes resulting from human errors”
- Should either be “this factor” or “these factors” – can’t quite tell from the context, but “this factors” is not correct
- Line 77 – “dimenstions”
- Spelling
- Line 168 – “a total number of76 items”
- Typo
Author Response
Dear reviewer, we thank you for the time you put into reviewing our document. We think your comments have helped us improve the final version, which you will find attached. You will find our answers to your second review comments in the attached document.
